# The Guadalfeo Monitoring Network (Sierra Nevada, Spain): 14 years of measurements to understand the complexity of snow dynamics in semiarid regions

María J. Polo[1], Javier Herrero[1], Rafael Pimentel[1], María J. Pérez-Palazón[1]

[1]Fluvial Dynamics and Hydrology Research Group - Andalusian Institute for Earth System Research (IISTA), University of Cordoba, Córdoba, 14071, Spain

*Correspondence to*: María J. Polo (mjpolo@uco.es)

**Abstract.** This work presents the Guadalfeo Monitoring Network in Sierra Nevada (Spain), a snow monitoring network in the Guadalfeo Experimental Catchment, a semiarid area in southern Europe representative of snowpacks with highly variable dynamics on both the annual and seasonal scales, and significant topographic gradients. The network includes weather stations that cover the high mountain area in the catchment and time-lapse cameras to capture the variability of the ablation phases on different spatial scales. The data sets consist of continuous meteorological high frequency records at five automatic weather stations located at different altitudes ranging from 1300 to 2600 m a.s.l. that include precipitation, air temperature, wind speed, air relative humidity, and the short- and long-wave components of the incoming radiation, dating from 2004 for the oldest station (2510 m a.s.l.) (https://doi.pangaea.de/10.1594/PANGAEA.895236); additionally, daily data sets the imagery from two time-lapse cameras are presented, with different scene area (30x30 m, and 2 km$^2$, respectively) and spatial resolution, that consist of fractional snow cover area and snow depth from 2009 (https://doi.org/10.1594/PANGAEA.871706), and snow cover maps for selected dates from 2011 (https://doi.pangaea.de/10.1594/PANGAEA.898374). Some research applications of these datasets are also included to highlight the value of high resolution data sources to improve the understanding of snow processes and distribution in highly variable environments. The datasets are available from different open source sites and provide both the snow hydrology scientific community and other research fields, such as terrestrial ecology, riverine ecosystems or water quality in high mountains, with a valuable and high-potential information in snow-dominated areas in semiarid regions.

## 1 Introduction

The warming trends in climate cause an alarming change in snow patterns over mountainous areas (Liston and Hiemstra, 2011). The dominant role of snowfall over these areas is being shifted to a rainfall-snowfall regime, which changes the snow accumulation dynamics (Musselman et al., 2017). Evaposublimation and snowmelt have also altered their behavior, with a clear affection on the hydrological response and changes in both the timing and availability of water resources. Taking into

account that one-sixth of the total Earth's population depends on snowmelt for water resources, i.e. hydropower production, drinking water and irrigation (Barnett et al., 2005), these areas require a particular scientific focus. Big efforts have been done in mountain hydrology in big Mountainous Ranges (Mankin and Diffenbaugh, 2015; Mernild et al., 2017) specifically those located in colder latitudes (Marty et al., 2017; Verbyla et al., 2017; Rizzi et al., 2018). However, there are a significant number of mountainous areas situated in mid-latitudes, warmer and drier, where snowmelt also constitutes the main component of the total water resources during the summer season. This is the case of semiarid/Mediterranean mountainous areas (Fayad et al., 2017), for instance, Sierra Nevada Mountain (Southern Spain) (Perez-Palazón et al., 2018), the eastern part of The Pyrenees (Spain) (Lopez-Moreno et al., 2009), Atlas Mountain Range (North Morocco) (Marchane et al., 2015), Mount Etna and Southern Calabrian Mountains (Italy) (Senatore et al., 2011), the southern face of The Alps (France and Italy), Taurus Mountain (Turkey), Mount Lebanon and Anti Lebanon (Lebanon) (Mhawej et al., 2014), The Sierra Nevada (USA) (Molotch and Meromy, 2014), and The Andes (Chile) (Váldés-Pineda et al., 2014). Moreover, the majority of them comprise spots or areas with high environmental value, as Biosphere Reserves in many cases with an outstanding number of endemism. Thus, these regions also constitute natural laboratories where warming trends are not only being noticed but also have significant impacts on the snow dynamics. Consequently, they are locations where these effects can be early evaluated provided the availability of key observations to analyze the major drivers of change (López-Moreno et al., 2017). A large number of current studies are focused on reproducing the most likely future climate scenarios, and their impacts, and trying to establish strategies to adapt to and mitigate these changes (Andrew et al., 2017; Ergon et al., 2018). However, the intrinsic variability of climate and the large spatial coverage of the snowpack in these areas, with a frequently patchy behaviour not only at the end of the snow season, but also during the fall and winter due to the different accumulation-ablation cycles, in Mediterranean mountain regions pose an opportunity to study the impact of the climate trends on the snow occurrence and persistence, and anticipate to these likely changes in higher latitudes.

Snow distribution in warm mid- and low-latitude locations is more variable and irregular than in northern regions: 1) several accumulation-melting cycles occur throughout the year, with very different duration (Pimentel et al., 2017c); 2) a wide range of snow depth states is usually found, generally closer to the order of magnitude of the surrounding microrelief (1-1000 mm) (Anderton et al., 2004); and 3) a particular patched snow distribution, ranging from one to hundreds of square meters (Pimentel et al., 2017a), is one of the major consequences of extremely variable conditions over these regions (i.e. high level of solar radiation between precipitation events, mild winters followed by cool springs and vice versa, and the alternance of severe drought periods and highly torrential wet seasons). All these variability sources make it necessary an accurate, continuous and well distributed monitoring network that is able to capture the main processes related to snow dynamics, both meteorological forcing and snow properties, and ultimately their impact on the river flow. However, the difficult access to mountainous areas complicate the installation and maintenance of meteorological stations, and somehow limits in practice to carry out continuous surveys and field campaigns to monitor the snow properties. Hence, there is an usual lack of weather stations in high mountain areas, which increases in semiarid regions since the monitoring efforts are concentrated in the agricultural areas in mid and low altitudes.

Sierra Nevada (Spain) is a clear example of high mountain areas in a semiarid context. This 80-km long mountain range runs parallel to the shoreline of the Mediterranean Sea in southern Spain, with frequent snowfalls from November to April in the area above 2000 m a.s.l. that usually undergo several ablations before the summer. Its proximity to the sea favours the coexistence of Mediterranean and Alpine climates within just a 40-km distance. Its particular location and steep topographic gradient make Sierra Nevada to be recognized as one of the most important reservoirs of biodiversity in Europe, with presence of a large number of endemic species (Heywood, 1995; Blanca, 1996, Anderson et al., 2011), as well as the major ski resort in South Europe during the spring. The projections of future climate in this area (Pérez-Palazón et al., 2018) point out the enhancement of the variability of the snowfall regime, and the increased frequency of ablation cycles during the season. However, the weather network was restricted to the area below 1500 m a.s.l. until the beginning of the XXI century, mostly concentrated below 1000 m a.s.l., and a significant gap was found to not only estimate snowfall but also analyze temperature and precipitation regimes in the mountains. The southern face of Sierra Nevada is the headwater area of the Guadalfeo River Basin and constitutes the major water source for urban supply and high-value crop irrigation in the coastal areas downstream, where a big reservoir was built in 2002 to face drought periods and the increasing development of coastal tourism and facilities. This work presents the monitoring network implemented in Sierra Nevada since 2004 to study the snow dynamics in this area and its hydrological impacts. The Guadalfeo Monitoring Network comprises five automatic weather stations located above 1300 m a.s.l., mainly in the Guadalfeo River Basin, that had been gradually installed during 2004-2017; additionally, three time-lapse cameras complement the network and provide high resolution images of the snow cover area on different spatial scales from 2009. The Guadalfeo Network has been developed and maintained since then, by the Fluvial Dynamics and Hydrology Research Group from the Andalusian Institute for Earth System Research. The included data set consists of 13-yr series of weather variables (2004-2018), 7-yr series of snow observations from terrestrial imagery (2009-2016), and a number of snow cover maps from a second camera on selected dates (during 2011-2013), from which snow modelling (Herrero et al., 2009, 2011) weather variables analysis and mapping (Aguilar et al., 2010; Aguilar and Polo, 2011; Herrero and Polo, 2012) , snowmelt-evaposublimation partitioning (Herrero and Polo, 2016), subgrid scale effects of patchy snow conditions (Pimentel et al., 2015, 2017c) and other snow-related processes or impacts have been studied (Millares et al., 2009, 2014; Gómez-Beas et al., 2012; Gómez-Giráldez et al., 2014; Egüen et al., 2016; Pimentel et al., 2017a; López-Moreno et al., 2017; Algarra et al., 2019).

## 2. The Guadalfeo River Basin

The Guadalfeo River Basin is one of the five main watersheds whose headwaters belong to the Sierra Nevada Mountain Range in southern Spain (Fig. 1). Its location in the southwestern face of Sierra Nevada results in a predominant southern orientation, with high insolation rates during the year, and a proximity to the Mediterranean Sea, which makes this watershed a very particular environment due to the coexistence of a Mediterranean climate in the lower elevations and an Alpine climate in the headwaters of the catchment. This main south-facing changes in the western zones, where the higher hillsides are north-

oriented. Low-population areas and marginal agriculture (supported by snowmelt-fed historical irrigation systems) are found in the mountains, whereas the coast and the bottom valley are densely populated with the local economy based on tropical crop growing and tourism. The highest area of the catchment (above 1000-1500 m a.s.l.) is included in the Sierra Nevada National Park, also Biosphere Reserve. There are two reservoirs in the watershed: Béznar (56 hm$^3$, maximum capacity) and Rules (111 hm$^3$, maximum capacity). The first one located in the Ízbor River, a tributary to the Guadalfeo River whose flow comes from the northern oriented area within the catchment, and the second one is located in the Guadalfeo River main stream, and collects water from all the southern face and the outflow from the Béznar reservoir. Additionally, small dams for hydropower generation are present upstream the reservoirs in the snow-dominated areas in the mountains. Rules dam location is selected as output of the watershed, since the flow regime is modified downstream and the available records cannot then be used to reproduce the natural fluvial regime; moreover, the daily inflow data to the reservoir are used for the closure of the water and energy balance on the catchment scale.

The Guadalfeo Experimental Catchment comprises the contributing area to the Rules dam, control point of inflows to the reservoir. This upstream catchment has an area of 1058 km$^2$, with an average elevation of 1418 m a.s.l, ranging from 300 to 3479 m a.s.l.  The average annual precipitation is very variable, even in consecutive years, and ranges from approximately 300 to 1200 mm in dry and wet years, respectively. The average daily temperature in the Guadalfeo basin  for the last 40 years is 12.4 ºC (Pérez-Palazón et al., 2015), which is higher than values recorded in other alpine climates with similar altitudes. These average values are calculated over a reference period (1690-2000) with all meteorological information available in the catchment for this period (red and black dots in Figure 2). The vegetation in the snow dominated area, mainly above 2000 m a.s.l., is composed by pastures and low shrubs (*Hormathophylla spinosa, Genista versicolor* and *Festuca Clementei*). Some isolated spots of reforested conifers can be also found in this area, but never in elevations higher than 2500 m a.s.l. On the contrary, out of the snow area, small oak, poplar and chestnut trees forests and traditional agriculture are the dominant areas.

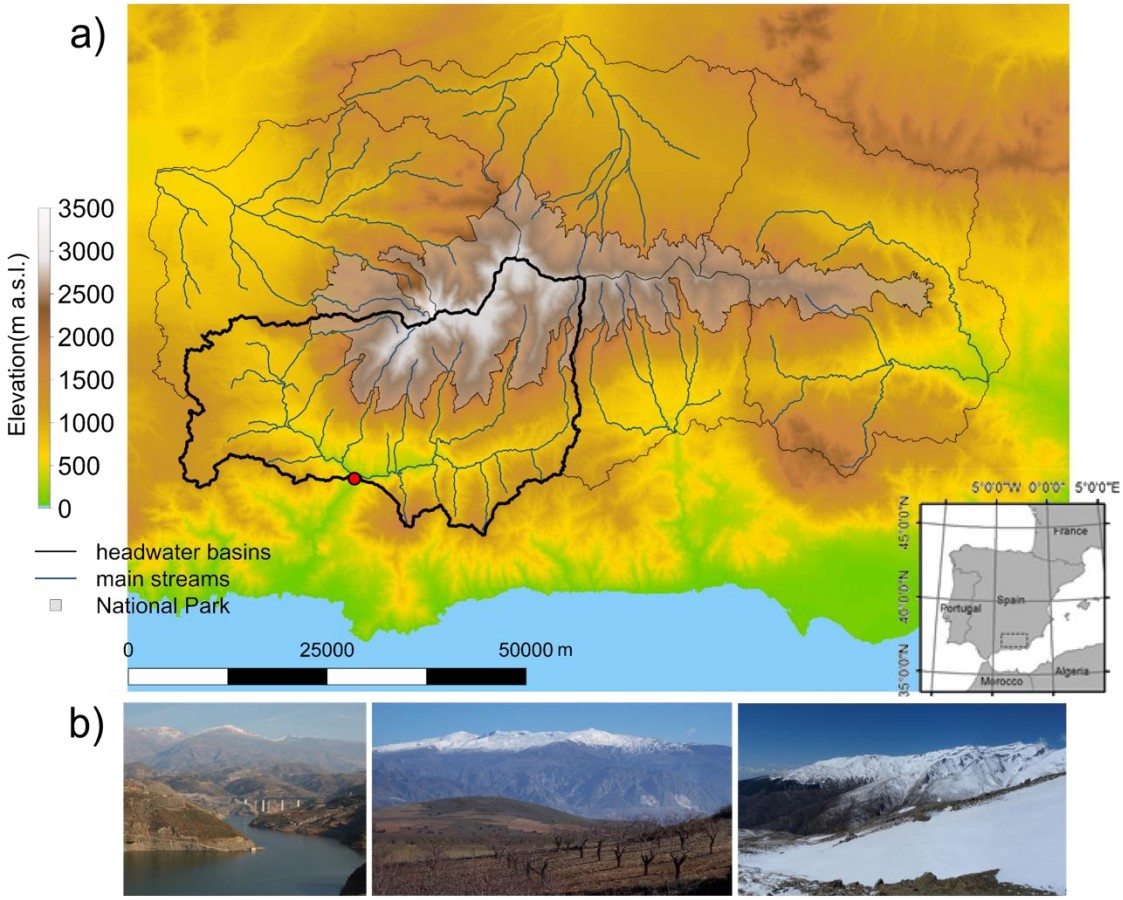

**Figure 1: The Guadalfeo Experimental Catchment study site in Sierra Nevada (Spain). a) Right, location in Spain. a) Left, topography of the area, major fluvial streams, and limits of the catchment (black line); the red circle corresponds to the Rules dam, closure of the catchment, and the greyish area is the National Park domain. b) Some representative views.**

The monitoring resources in the Guadalfeo Experimental Catchment consist of: the Guadalfeo Network (Fig. 2), whose weather and camera stations are mostly located in this catchment, all of them above 1200 m a.s.l.; other weather station networks operated by public national and regional agencies (mostly below 1000 m a.s.l.); the Refugio-Poqueira experimental site (surrounding area of PG2 in figure 2) at 2500 m a.s.l. where snow and soil are monitored by additional instrumentation, and systematic field campaigns for snow monitoring are carried out; gauge stations in the main streams and inflow estimations to the reservoirs, both operated by the Water District of South Mediterranean Catchments.

## 3. The Guadalfeo Network

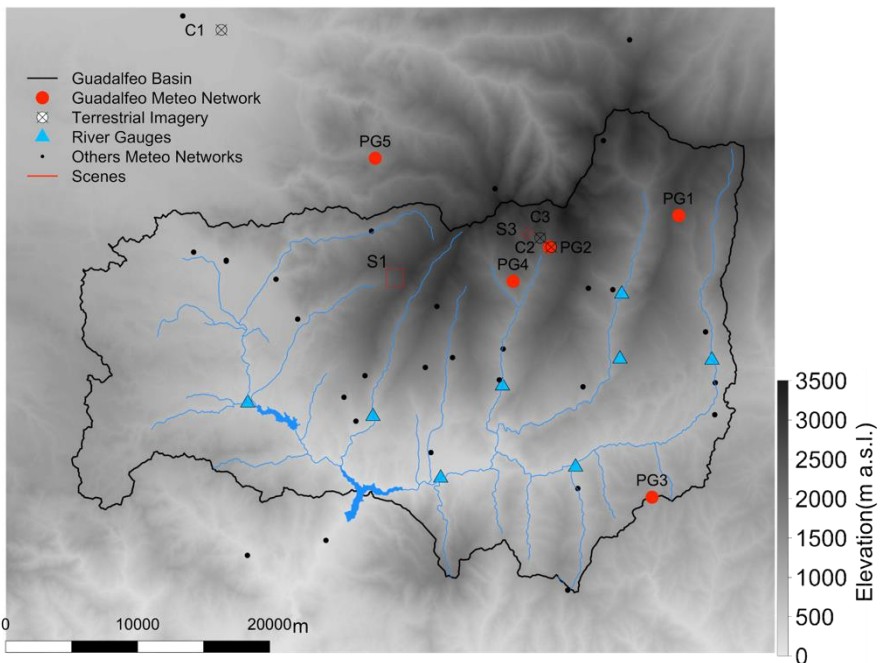

**Figure 2: The Guadalfeo Network in Sierra Nevada (Spain). PG, weather stations (red circles); C, time-lapse cameras (cross-circles); S, the camera scenes (red rectangles; S2 is a near-field scene and cannot be represented due to its size). Black points identify the location of weather stations belonging to other meteorological networks in the Sierra Nevada area. The limits of the Guadalfeo Experimental Basin (black line), the main river streams (blue line), gauge stations (blue triangles) and reservoirs (blue areas) are included over the elevation map of the area.**

The Guadalfeo Network was specifically designed to both fill the gap in the weather station distribution above 1000 m a.s.l. in the Sierra Nevada mountains and monitor the weather and snow regime in the snow-dominated areas. Figure 2 shows the location of the automated weather stations (PG points) and camera locations for terrestrial imagery acquisition, including their respective associated scenes, together with the reservoirs and gauge stations in the river network. The weather stations were sequentially installed since 2004 and they are equipped to monitor the radiation components. The following points describe the network in detail.

### 3.1 Weather stations

**Table 1. Main characteristics of the weather stations in the Guadalfeo Network; elevation (m a.s.l.), date of installation, instruments, and in situ pictures.**

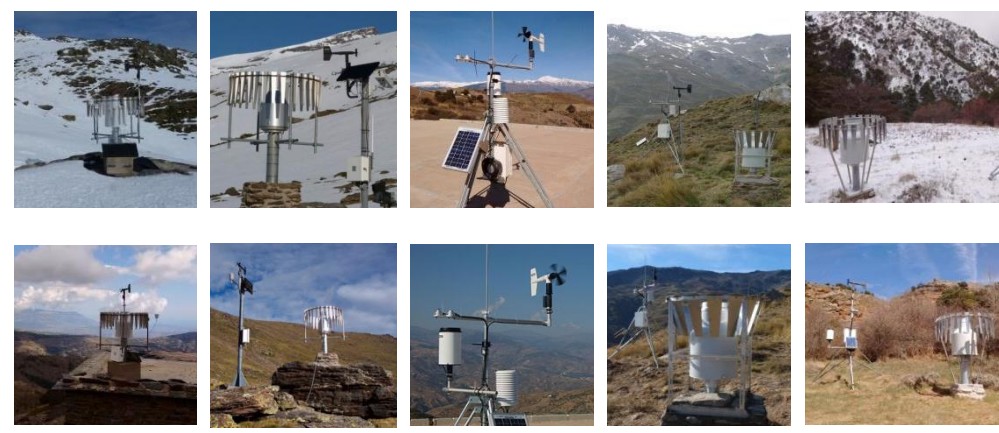

| ID | PG1 | PG2 | PG3 | PG4 | PG5 |
|---|---|---|---|---|---|
| **Elevation** | 2470 | 2510 | 1332 | 2141 | 1675 |
| **Installation** | Nov 2005-operational | Nov 2004-operational | Aug 2009-operational | Oct 2012-operational | Mar 2017-operational |
| **Rain gauge** | OTT Pluvio 2 | Geonor T-200B | Young 52203 | OTT Pluvio 2 | OTT Pluvio 2 & Young 52203 |
| **Temperature Relative Humidity** | Campbell Scientific CS215 | Vaisala HMP45C | Vaisala HMP45C | Vaisala HMP45C | Campbell CS HC2S3 |
| **Pyranometer** | - | Kipp&Zonnen SP Lite | Hukseflux LP02-05 | Hukseflux LP02-05 | Hukseflux LP02-05 |
| **Pyrgeometer** | - | Kipp&Zonen CGR3 | Hukseflux IR02 | Hukseflux IR02 | Hukseflux IR02 |
| **Alpine Wind monitor** | Young 05103-45 | Young 05103-45 | Young 05103-45 | Young 05103-45 | Young 05103-45 |
| **Barometer** | CS100, Setra 278 | Druck RPT410F | CS100, Setra 278 | CS100, Setra 278 | CS100, Setra 278 |

Five weather stations are currently included in the Guadalfeo Network (red dots in Figure 2). Table 1 and Figure 3 show, respectively, the equipment installed in each station and the chronological sequence of the network development and data acquisition. Three domains can be identified in the stations distribution:

- PG1 (Nov, 2005) and PG2 (Nov, 2004) are the highest stations of the network (around 2500 m a.s.l.); they are located in two of the main headwater subbasins within the catchment, in the Bérchules and Poqueira valleys, respectively. PG4 (Oct, 2012) is also located in the Poqueira river subbasin at a lower elevation than PG2 to capture the altitudinal gradient, which is particularly important in this valley due to the steepness of the terrain.

- PG3 (Aug, 2009) is located outside of Sierra Nevada Mountain Range, in the left margin of the Guadalfeo river in the Contraviesa Mountain, exactly in the crest of the range; this area lacked stations to capture the altitudinal gradients of the meteorological variables, and this point was needed to improve the precipitation distribution and, thus, the

hydrological modelling of the inflows to the Rules reservoir, that also receives runoff from this area. Moreover, sea influence is also relevant on this location.

- Finally, PG5 (Mar, 2017) is so far the only station in the network that is located out of the Guadalfeo Experimental Catchment; its location was chosen as a first step to fill the gap of meteorological information in the northwestern
area of Sierra Nevada.

All the stations monitor six weather variables: precipitation, temperature, relative humidity of the air, incoming shortwave radiation (not measured by PG1), wind and atmospheric pressure (Table 1). The incoming longwave radiation is additionally measured in PG2, PG3 and PG5. At each station, a data-logger records 5-min averages of 5-s sampling rate observations; data are transmitted via modem to the Andalusian Institute for Earth System Research where they undergo a quality-control process.

This quality-control consists of a standard limit checking, intercomparison of values for each variable with the nearby stations, and internal checking of the values of a given variable in the context of the whole set of information at the same station (including video image). As a result, any suspicious 5-min record is removed from the series; to generate hourly series, only periods with less than two removals are kept (i.e. a maximum uncertainty of 8.33% is due to gaps in the record), and only days with complete hourly values are included in the daily series. No gap filling is done on the resulting series on any time scale to

keep the original data set, and it is up to the user to apply any gap filling technique if required. Additionally, periodic calibration of the solar radiation, temperature and relative humidity sensors is performed on an annual basis by means of portable calibrated sensors provisionally installed beside each station. The fraction of removed records varies depending on the variable, but is generally below 0.01% in the network.

No correction is done on the data to account for potential snow undercatch; the pluviometers above 1500 m a.s.l. are equipped

with either Alter or Tretiakov shields to improve snow catching; the different field tests performed at the stations sites showed negligible undercatch amounts during the most frequent events. The mean value of wind speed during snowfall events ranges between 3.1 and 4.8 m/s at the stations PG1 to PG4 (with mode values ranging from 1 to 2.5 m/s), and 0.8 m/s at station PG5 (with mode value of 1 m/s). However, under extreme conditions this might not hold; during the study period the maximum wind speed registered ranged from 12.1 m/s (PG3) and 24.7 m/s (PG1). Again, the data are delivered without any modification

so that the interested users can decide whether correct or not for their own applications, provided that the wind data are also included in the dataset.

With very few exceptions, the stations have been recording data on a continuous basis since their deployment; PG1 had some initial problems and was re-installed at the end of summer 2009 (Figure 3).

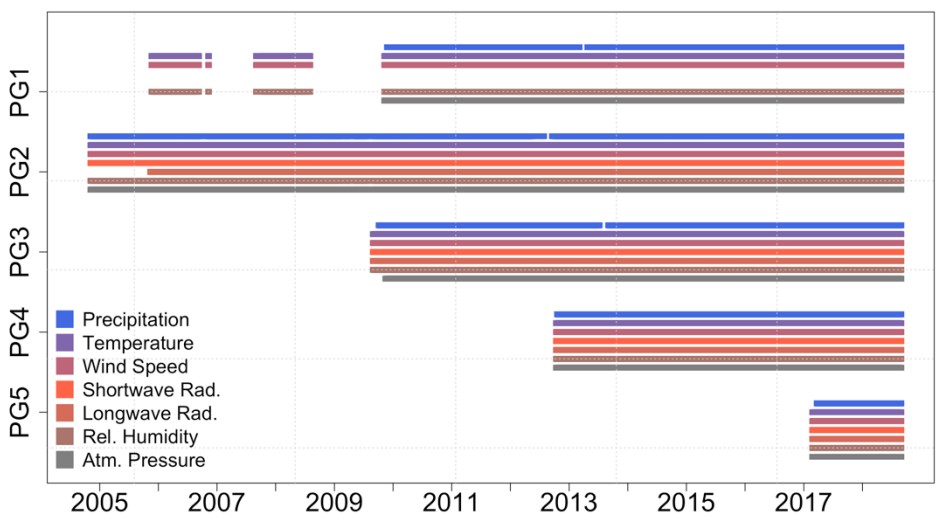

**Figure 3. Recording period and data availability for each weather station and variable in the Guadalfeo Network (Sierra Nevada, Spain) since 2004.**

### 3.2 Terrestrial imagery

Currently, the Guadalfeo Network includes three time-lapse cameras (Fig. 2), all located above 1200 m a.s.l., which cover different scenes associated to different spatial scales. Table 2 shows selected characteristics of each camera together with an example image to illustrate the scene they capture. Two of them (C2 and C3) are both located and cover a scene in the highest region of the Guadalfeo Experimental Catchment, but with different spatial scale and resolution.

-   C2, at the Refugio Poqueira experimental site in the surroundings of PG2, covers a 30x30m scene (equivalent to the
spatial resolution of the Landsat TM sensors) and is devoted to microscale effects on the snow ablation; it also complements the information recorded by the weather station, since it provides sub-daily measurements of snow depth (by means of calibrated rods easy to identify from the images) and estimations of snow-covered area in the scene (after processing of the images).

-   C3 covers a larger scene (500x500m) and provides subdaily estimations of snow cover area on the hillslope scale, as
a validation data source for the results from C2.

-   Finally, C1 is focused over a larger hillslope in the only subbasin of the Guadalfeo Experimental Catchment in the northern face of Sierra Nevada; this camera covers the largest scene (2 km$^2$) from the network and the images provides both monitoring the snow cover area evolution on this area, and ground-truth data for the validation of and/or filling the snow cover area maps from satellite sources.

**Table 2. Main characteristics of the time-lapse cameras in the Guadalfeo Network; elevation (m a.s.l.), date of installation, instruments, frequency and spatial resolution, and in situ pictures of both their locations and scenes.**

| ID | C1 | C2 | C3 |
|---|---|---|---|
| Name | Caballo hillside | Refugio Poqueira | Veleta-Carihuela |
| Camera | Campbell CC5MPX | CC640 Campbell Scientific | MOBOTIX M25 |
| Installation | 2011/11/20 – operational | 2009/07/22- operational | 2011/12/09-operational |
| Temporal Resolution | 13 images per day | 5 images per day | 13 images per day |
| Spatial scale | Hillside (~2 km) | Detail (~30 m) | Hillside (~500 m) |
| Photo resolution | 2592x1944 pixels | 640x504 pixels | 3000x2000 pixels |

The terrestrial images are processed in a two-step methodology: georeferencing and color detection. Georeferencing uses a digital elevation model (DEM) to provide the image with spatial coordinates by means of a standard automatic computer vision algorithm (Foley, 1996; Fiume, 2014). Secondly, for the detection of snow-pixels, a clustering K-means algorithm is applied (MacQueen, 1967) that classifies the pixels in the image into two clusters: snow-covered and snow-free pixels (Pimentel et al., 2015); finally, the snow cover area in the image is calculated by aggregation of pixels in each cluster. Additionally, the snow

depth, h, was obtained in each image using the red-painted poles installed in the scene area and using the same clustering algorithm to determine pole and no-pole pixels in a predefined searching window that isolates the poles in the images (Pimentel et al., 2015, 2017a). The accuracy of the observations from the camera data was estimated as 0.075 $m^2$ $m^{-2}$ for the snow cover fraction (SCF) and 50 mm for the height of snow (HS). The associated error for SCF is due to the combined effects of both the georeferencing process and the snow detection algorithm; the former depends on the relationship between the number of

pixels in the images and the resolution of the local DEM, and the latter is related to the accuracy of the K-means algorithm used. In the case of HS, the error depends on the position of the rods installed in the control area and also on the accuracy of the clustering algorithm used.

## 4. Data description

### 4.1. Meteorological data from the weather stations

Figure 4 shows the temporal evolution of the meteorological data registered in each station of the Guadalfeo Network during the period 2004-2017 on an annual basis, together with their seasonal distribution within the year by means of their monthly

descriptor during the monitoring period at each site. The different graphs illustrate relevant features of the climate regime in high mountain areas in semiarid regions, as expressed in the Introduction section: i) the high variability of the precipitation and temperature regimes on different time scales, and the seasonality of the precipitation pattern; ii) the larger temperature interval on a local basis, due to the spring and summer periods, much warmer than other snow domains in higher latitudes; iii) the importance of the radiation components, with higher amounts of incoming radiation on an annual level, especially during spring-summer periods; iv) the influence of the proximity to the sea, with air relative humidity annual intervals rather stable during the study period for a given station.

The distribution of the different stations in the network allows some spatial comparison, despite the different duration of the time series at each site (Fig. 3). PG5 is the only station on the northern face of Sierra Nevada and it has only one year of records; PG1, PG2 and PG4 are located on the southern face, and PG3 is on the southern water divide of the Guadalfeo River Basin (out of the snow-dominated area in the watershed). PG2-PG4 trace a North-South gradient in the same valley, whereas PG2-PG1 trace a West-East gradient along the same altitudinal line (Fig. 2); their respective annual temperature intervals and descriptors mainly reflect the altitude level within the Sierra Nevada range, and the mean monthly descriptors show the clear seasonal pattern in this latitude.

Regarding the precipitation regime, PG2-PG4 show the maximum altitudinal band in the southern face around 1900-2200 m a.s.l., already identified in other works (Pérez-Palazón et al., 2018) and the decline of precipitation along the West-East axis (PG2-PG1) associated to the major trajectories of the Atlantic cyclonic fronts in this region. Fall and late winter-early spring are the seasons concentrating most of the precipitation, being November and March the most variable months; very scarce precipitation is found from June to August at all sites.

Wind speed can reach high values during the snow season in the snow-domain sites (PG1, PG2, PG4), which has a clear impact on evaposublimation rates and snow redistribution in these areas (Herrero and Polo, 2016).

The local incoming radiation components reflect the annual evolution of the solar cycle, with the minimum solar activity reported during 2007-2009 and a minimum around January 2009 (NOAA, 2016), which corresponds with the annual incoming shortwave radiation in 2008-09 in Fig. 4 (1 Sep 2008-31 Aug 2009); the incoming longwave radiation also shows this regime, being one order of magnitude lower than the shortwave component on an annual basis. The altitudinal gradient among the different stations can also be observed from the annual values of both components: the shortwave annual value increases with height on the southern face of Sierra Nevada (PG4-PG2), and PG3 and PG5 show the influence of their location (southern position in the network, and northern face of Sierra Nevada, respectively), whereas the longwave annual value reflects the different temperature range, as expected. The seasonal distribution follows a clear pattern during the year, with the minimum values in December and February for the short- and longwave components, respectively, as expected.

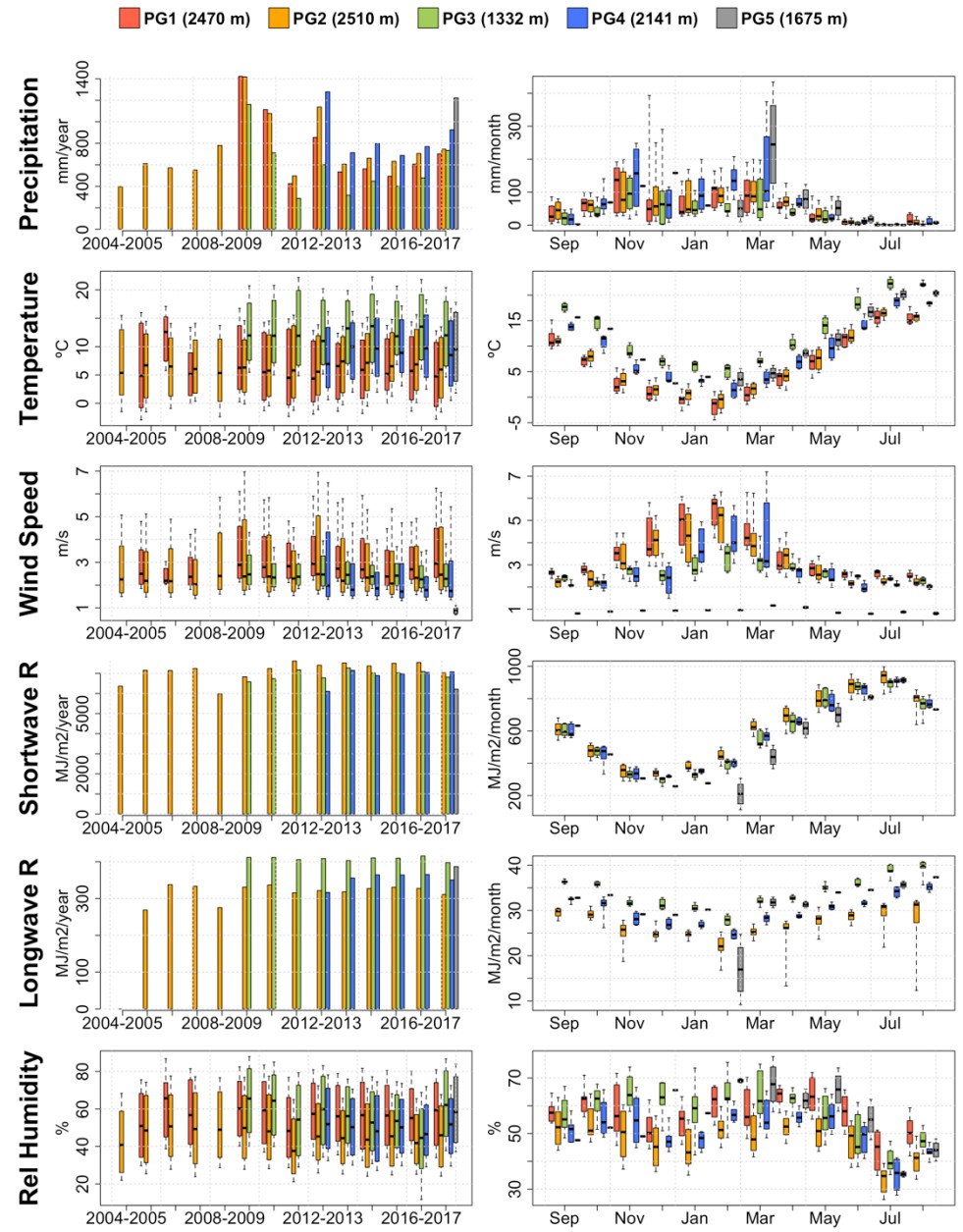

**Figure 4: Annual and monthly descriptors of the meteorological variables monitored during 2004-2017 by each weather stations in the Guadalfeo Network. Left: annual precipitation, incoming short- and longwave radiation; annual mean daily temperature, wind speed and air relative humidity. Right: mean monthly precipitation, incoming short- and longwave radiation; mean monthly temperature, wind speed and air relative humidity. Box-plots show the median values (black line), the interval range and the 10th and 90th percentiles (whiskers) of each sample (series duration in Figure 3).**

Finally, the air humidity annual values show the influence of the semiarid context of this mountain range, with intervals range larger than those usually found in snow domains in higher latitudes; both the precipitation regime and vicinity to the sea influence the local values found at each site. The seasonal pattern also reflects the seasonal pattern of precipitation, with fall and late winter-early springs concentrating the local maximum values and the largest variability; the extremely dry conditions in July can also be observed.

## 4.2 Snowpack variables from terrestrial imagery

Figure 5 shows selected images from C1 and C2 cameras in the Guadalfeo Network during the period 2009-2016, together with the resulting snow maps in each scene for different dates.

As described in section 3, C2 is implemented in the experimental site in Refugio Poqueira and the 30x30 m scene it covers corresponds to the surrounding area of the weather station PG2, at 2510 m a.s.l. This is the longest time series of images in the network, and its spatial resolution and scale allows for monitoring snow depth. The 7-yr time series of both fractional snow cover (FSC) in the scene area and snow depth on a daily basis is represented in Figure 6 for each year in this period, and Table 3 includes some statistical annual descriptors of both variables during the snow season (1 Nov-31 May).

The results clearly show the extremely high variability of the snow cover fraction in this site, and the relevance of the ablation processes during the snow season, with null values for the mode of FSC and snow depth all the years, with the exception of 2010-2011, which exhibited a deep and persistent snowpack during the season (Figure 6). The variation coefficient ranges between 0.73 and 1.63 around a global value of 1 for the study period, being lower or close to 1 for the years with longer persistence of the snowpack in the site, and higher than 1 for lower persistence conditions.

The snow depth shows a higher variability, with variation coefficient values ranging between 0.98 and 2.47, around a global value of 1.71 for the study period, and the same pattern described for FSC.

2009-2010 and 2010-2011 are representative years of persistent snowpacks from fall to spring, even though several complete ablations took place during the snow season. On the other hand, other years show a clear predominance of the snow-free periods. This annual variability results in highly significant differences in the annual snowmelt fraction (and volume) feeding aquifers and surface waters.

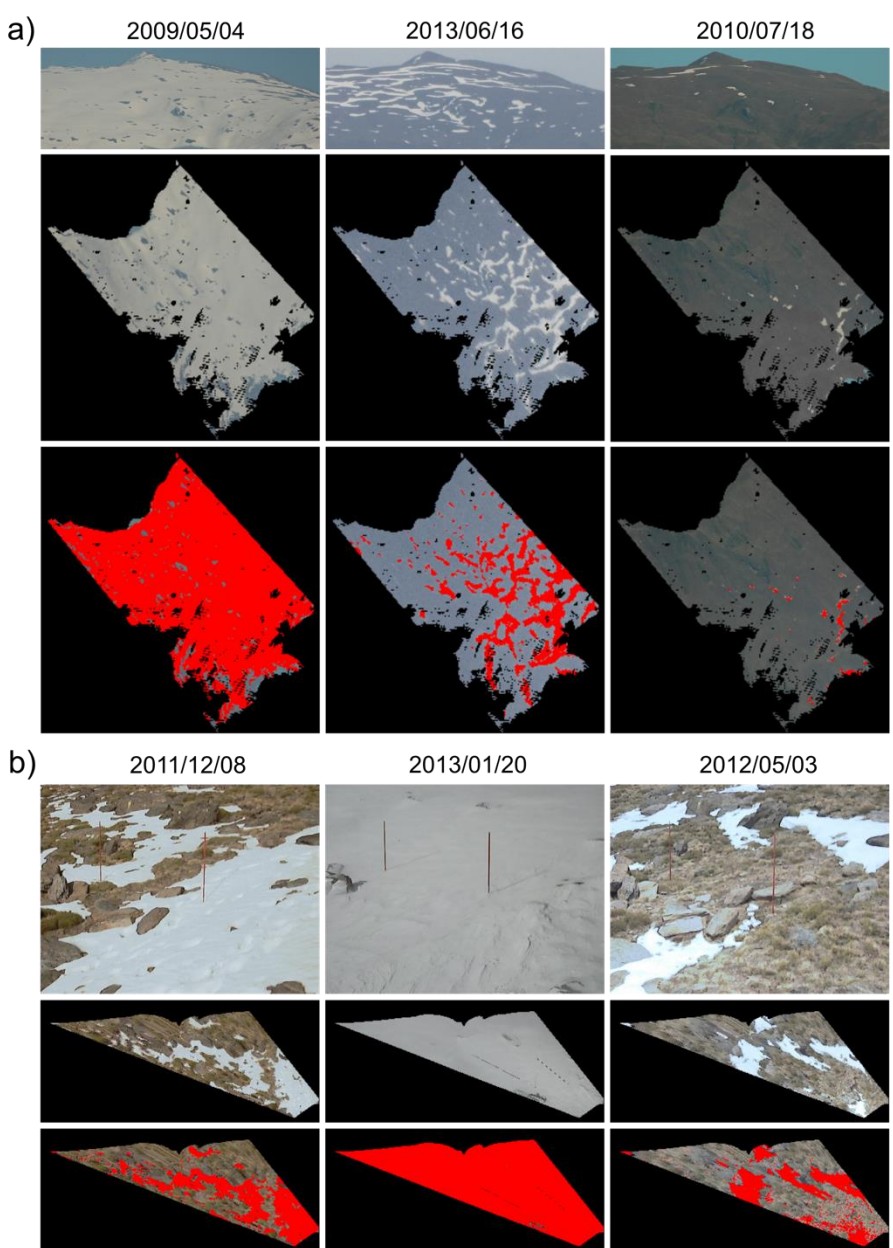

**Figure 5: Selected examples of the images, georeferenced projections and snow maps (red pixels) from the time-lapse cameras a) C1 (2 km2) and b) C2 (30x30m) in the Guadalfeo Network that illustrate the patchy pattern of the spatial distribution of the snow in the study site.**

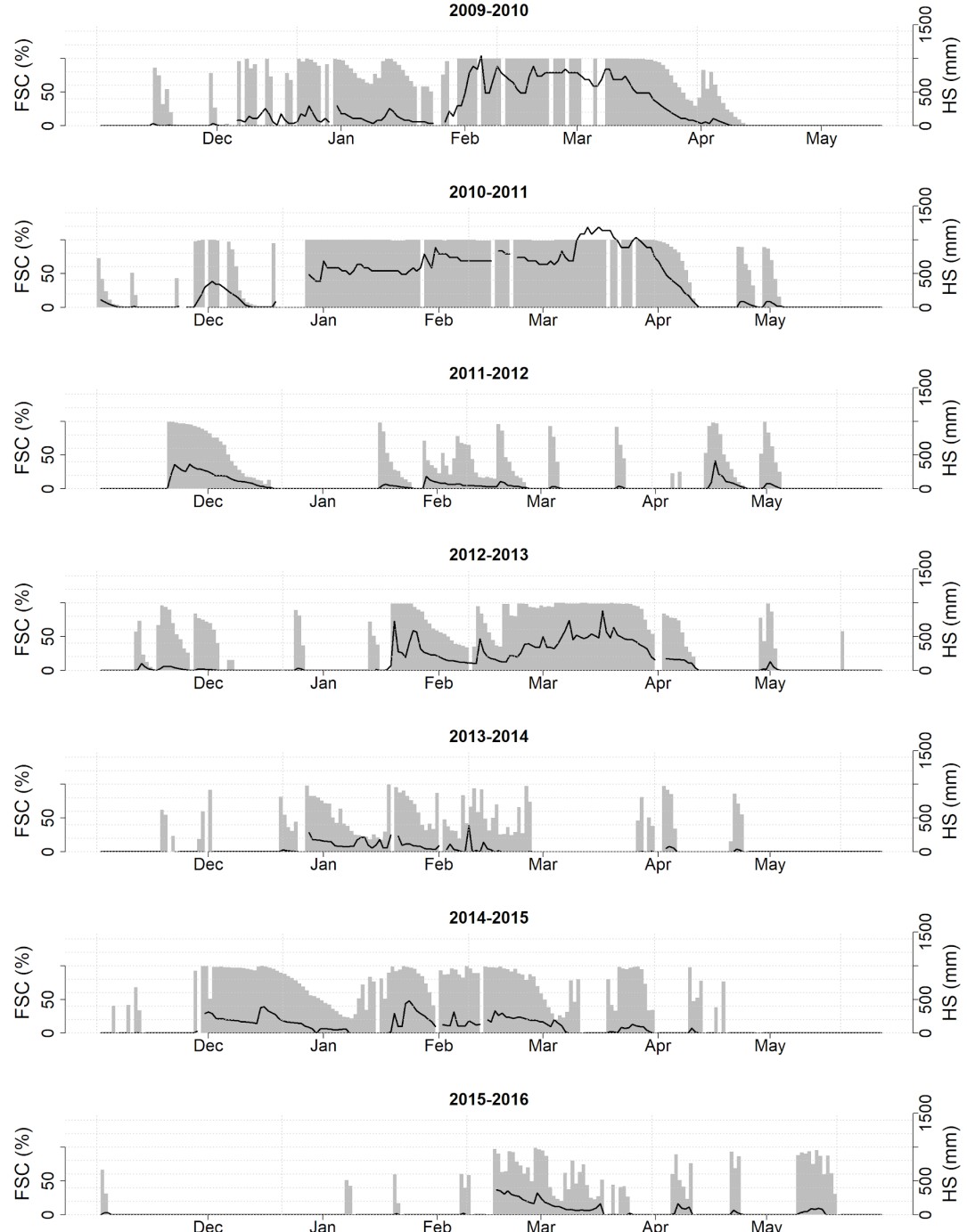

**Figure 6: Snow variables retrieved from the images of the time-lapse camera C2 in the Guadalfeo Network. Evolution of the observed daily fractional snow cover (FSC) and height of snow (HS) from the 30x30m scene during the snow season (1 Nov-31 May) during each hydrological year of the period 2009-2016.**

**Table 3. Selected statistical annual descriptors of the daily fractional snow cover (FSC) and height of snow (HS) during the snow season (1 Nov-31 May) in the C2 scene at the Refugio Poqueira experimental site in the Guadalfeo Experimental Catchment (located by PG2, 2510 m a.s.l., Figure 2); std, standard deviation.**

| | FSC (%) | | | | HS (mm) | | | |
| --- | --- | --- | --- | --- | --- | --- | --- | --- |
| | mean | mode | median | std | mean | mode | median | std |
| 2009-2010 | 0.54 | 0 | 0.68 | 0.43 | 220 | 0 | 82 | 291 |
| 2010-2011 | 0.62 | 1 | 0.99 | 0.45 | 375 | 0 | 341 | 367 |
| 2011-2012 | 0.23 | 0 | 0.00 | 0.33 | 42 | 0 | 0 | 81 |
| 2012-2013 | 0.42 | 0 | 0.35 | 0.42 | 129 | 0 | 11 | 191 |
| 2013-2014 | 0.28 | 0 | 0.18 | 0.33 | 33 | 0 | 0 | 66 |
| 2014-2015 | 0.44 | 0 | 0.41 | 0.42 | 80 | 0 | 0 | 110 |
| 2015-2016 | 0.19 | 0 | 0.00 | 0.31 | 30 | 0 | 0 | 74 |
| TOTAL | 0.41 | 0 | 0.29 | 0.42 | 140 | 0 | 11 | 239 |

On the other hand, in the study period, three different years are found with scarcity of snow: 2011-2012, 2013-2014 and 2015-2016, with similar order of magnitudes for both FSC and snow depth on an annual basis. However, significant differences are found in their respective seasonal pattern of snow persistence. From Figure 6, it can be easily observed that fall, winter and late winter-early spring, respectively, are the snow persistence seasons in each of these three years, whereas winter, spring and fall, respectively, concentrate the highest number of days without snow in this site. This seasonal variability has an expected impact on the pattern of the ablation phases (Pimentel et al., 2017c) and the timing of the peakflows of snowmelt, but also on the partitioning of snowmelt/evaposublimation fluxes from the snowpack (Herrero and Polo, 2016) that finally determines the available water volume input to the water systems in the basin and the fluvial recession flows (Aguilar and Polo, 2016).

**4.3 Data applications for research and operational capabilities**

The presented data have been used in different research works on the snow dynamics in Mediterranean regions and as input data sources to operational applications of these results. Figure 7 shows the importance of snowfall at each station in the Guadalfeo Monitoring Network on both an annual and monthly basis (Pérez-Palazón et al., 2018). The snowfall/rainfall classification is calculated on an hourly basis by using a threshold temperature of 0ºC at each station.

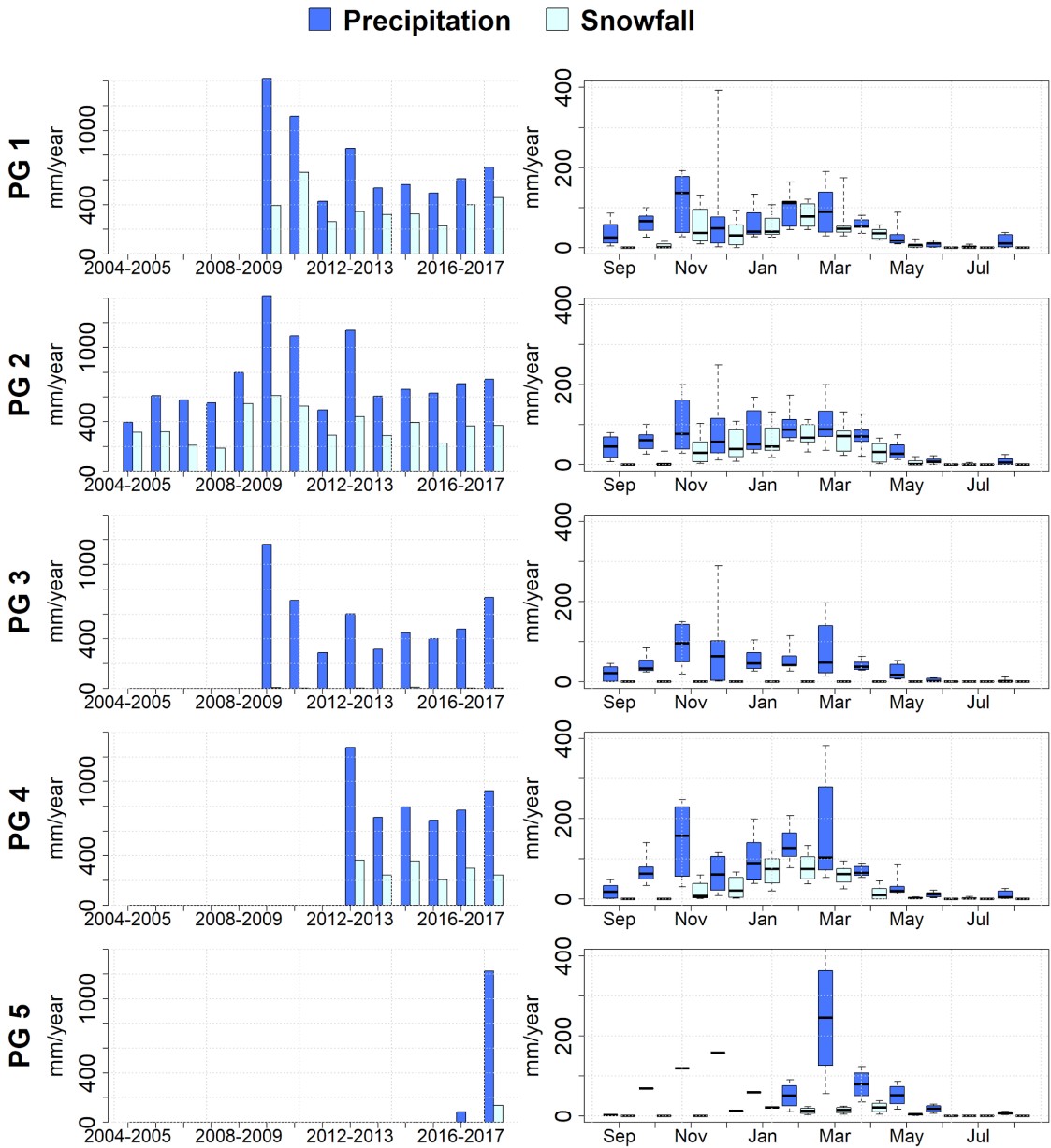

**Figure 7: Precipitation and snowfall in the weather stations of the Guadalfeo Network in Sierra Nevada (Spain). Left: Annual precipitation and snowfall at each station during the 2004-2016 period. Right: Mean monthly precipitation and snowfall at each station during the study period; box-plots show the mean values (black line), interval range and the 10th and 90th percentiles (whiskers) of each sample (series duration in Figure 3).**

The altitudinal gradient of precipitation does not directly correlate with that associated to snowfall. PG1 and PG2 show the highest values of annual snowfall, with a similar monthly pattern and volumes; PG4 registers higher precipitation amounts but

results in lower snowfall values, especially during fall. The short recording period of PG5 prevents any further analysis; as expected due to the location in the northern face of Sierra Nevada, the recorded annual precipitation is higher than its southern sister stations, but its occurrence mainly during the late winter-early spring decreased the amount of snowfall during this particular year. PG3 is not located in a snow domain area.

The snow season usually expands from November to April, with some occurrences of sometimes heavy snowfalls in May-June at the highest stations.

The monitoring of the incoming radiation components led to the validation of the conclusions of the first attempt to model the snowpack accumulation-ablation cycles in the Refugio Poqueira experimental site (Herrero et al., 2009). This work estimated the sublimation rates from an energy balance model, but no direct observations were available at that time. From the incoming
longwave radiation measurements at station PG2, an empirical expression for the atmospheric emissivity was derived (Herrero and Polo, 2012) and different field campaigns were done to measure and validate the estimated snowmelt/evaposublimation fluxes on a seasonal and annual scales at this experimental site (Herrero and Polo, 2016). The results highlight the importance of the evaposublimation fraction in mountain areas in semiarid regions, and the need for considering this component in the water balance in these catchments to avoid overestimations of the contribution of the snowpack to the water volumes in aquifers
and reservoirs from the observed snow cover area in the headwater zone.

The fractional snow cover and snow depth time series from the time-lapse camera imagery constitute a key information for studying the microscale effects on the snowpack ablation dynamics and the patchy pattern evolution. These effects need to be quantified in the distributed modelling of snow to prevent both over- and under-estimations of the snow water equivalent in
the snow pack when the energy and mass balance equations are solved over the cell area of a gridded representation of the snow cover area. The analysis of both the dynamics and timing during the season of the accumulation and ablation phases in the Refugio Poqueira experimental site (S2-C2, and PG2 in Figure 2) led to the retrieval of different depletion curves, that were associated to different antecedent conditions during the snow season so that a decision tree could be applied in the workflow of the model to better approximate the actual fractional snow cover and snow depth on a pixel basis (Pimentel et al.,
2017c). Figure 8 shows the dimensionless version of these depletion curves, with a single accumulation pattern but four potential patterns depending on the antecedent conditions during the snow season. Curve 0 describes the accumulation phase, which is initially very fast to slow down close to a 50% of the maximum snow cover area and reach a maximum snow depth threshold beyond which the area is completely covered. Curves 1 to 4 describe the ablation phase under different conditions: large amounts of snow from long accumulation phases, with very compact state and a high level of metamorphism (Curve 1),
or from short accumulation and non-persistent phases (Curve 2); lower amounts of snow accumulated during autumn-winter with longer snowmelt phases (Curve 3) or spring, with quick ablation due to the warmer conditions (Curve 4).

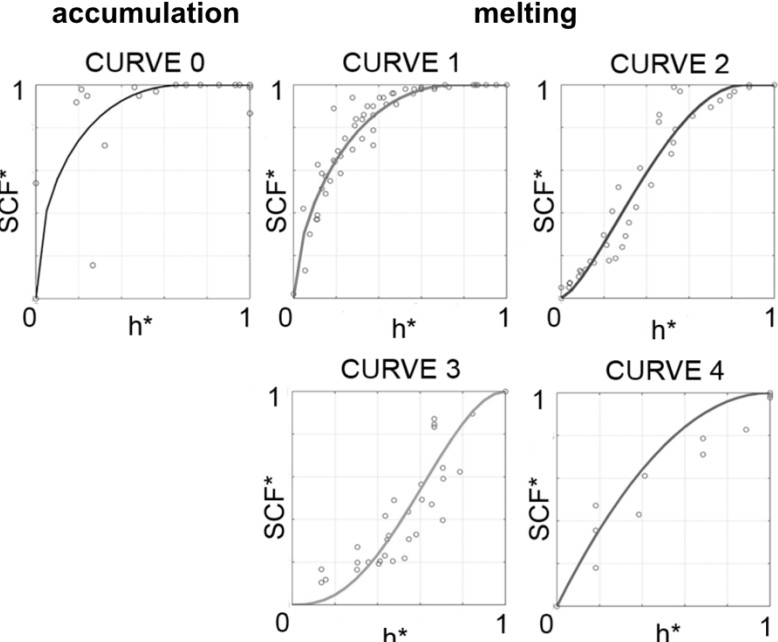

**Figure 8: Snow depletion curves on a cell basis retrieved from the data from the images of the time-lapse camera C2 at Refugio Poqueira in the Guadalfeo Network. Different accumulation (curve 0) and ablation (curves 1-4) patterns relate the dimensionless fractional snow cover (SCF\*) and snow depth (h\*) at the 30x30m scene during the snow season (1 Nov-31 May) from data during the 2009-2013 period. Adapted from Pimentel et al. (2017b).**

Finally, the data set from the cameras covering the hillslope scale scenes (C1 and C3) constitute valuable ground-truth information for the remote sensing community working on the retrieval of snow products. Figure 9 shows the analysis of the performance of two different algorithms for quantifying the snow cover area on large areas from Landsat TM images (Pimentel et al., 2017a) by means of their comparison with the snow maps of the 2 km² scene from C1 images (Figure 3 and Table 2). Additionally, to the expected conclusion about a fractional approach being more accurate than a binary algorithm for snow cover area estimations, it must be highlighted that the data allowed the quantification of the over- and under-estimation level by each approach for different states of the snowpack, together with the identification of states for which both algorithms provided similar results or failed in achieving an adequate approximation. Despite the current availability of high spatial resolution satellite data sources, such as the Sentinel missions, these cannot provide long time series to analyze trends and changes in the snow regime, being Landsat data the most adequate data source in terms of long series and high spatial resolution in these highly heterogeneous snow patterns domains.

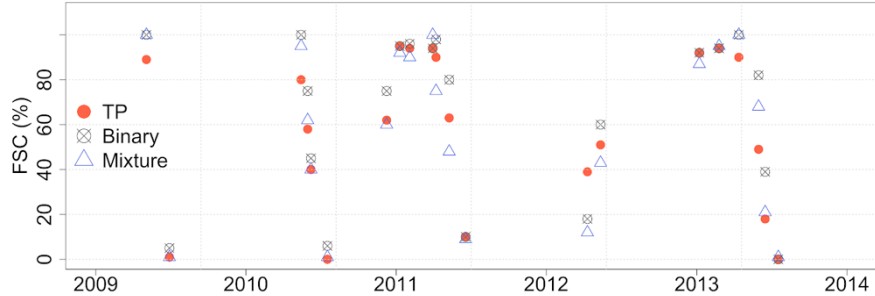

**Figure 9:** Comparison between the observed fractional snow cover (FSC) evolution (2009-2014) in the camera C2 scene area from the Guadalfeo Network (TP, red dots) and the estimated value from two different retrieval algorithms based on Landsat TM data: a binary approach (snow-no snow), and a fractional approach from a spectral mixture algorithm (FSC in every pixel of the satellite image). Adapted from Pimentel et al. (2017a).

## 5. Data availability

All data presented in this work are available from different open-access sources. The weather stations in the Guadalfeo Network are maintained by the Fluvial Dynamics and Hydrology Research Group of the Andalusian Institute for Earth System Research (University of Cordoba), and the data sets are included and updated in the CLIMA public database of the Andalusian Regional Government. The meteorological hourly and daily data sets are provided as .txt files from the following link, https://doi.pangaea.de/10.1594/PANGAEA.895236.

The fractional snow cover and snow depth data set at Refugio Poqueira site (camera C2) can be downloaded from https://doi.org/10.1594/PANGAEA.871706 (Pimentel et al., 2017b); the snow cover maps for selected dates from 2011 (camera C1) are available from https://doi.pangaea.de/10.1594/PANGAEA.89837 the snowmelt and evaposublimation fluxes referenced in section 4.3 are accessible from https://doi.org/10.1594/PANGAEA.867303 (Herrero et al., 2016).

The updated meteorological records and quasi-real time representation of the snow cover area in the Guadalfeo Experimental Catchment can also be accessed at http://www.uco.es/dfh/snowmed/ (in Spanish).

## 6. Final summary

This work presents the Guadalfeo Monitoring Network in the Guadalfeo Experimental Catchment, Sierra Nevada (southern Spain), an Alpine headwater area in a semiarid region with steep topography and sea-climate influence. The data sets include both meteorological time series from the weather stations in the network, and fractional snow cover area and snow depth time series retrieved from imagery from time-lapse cameras in selected points in the network. Different examples of research applications are also included to highlight the value of these research monitoring networks to acquire high resolution and quality data sets, absolutely essential for capturing the significant scales of the snowpack regime in these highly heterogeneous areas.

The data shown are representative of the main issues characterizing high mountain climate in semiarid catchments, and provided the research team with the basis for further understanding the major drivers of the snow accumulation-ablation patterns, the interaction of the microrelief or the partitioning of snowmelt and evaposublimation under different predominant conditions. The significant gradients of the snowpack regime found in Sierra Nevada both in the North-South and West-East axes still pose questions to be answered in a context of increasing variability of the climatic drivers in these regions, where the associated impacts not only on water availability during the dry season and years, but also on the operational design and exploitation of the water storage systems (reservoirs, aquifers…), constitute a priority issue from both the scientific and technical point of view.

The Guadalfeo Network is a live structure that keeps on expanding funded by different research projects and specific actions of infrastructure investment and maintaining. A new weather station has recently started recording data at 2500 m a.s.l. on the northern face of Sierra Nevada, and the Refugio Poqueira experimental site (PG2) has been equipped with additional sensors (snow water equivalent measurement, and a 4-component pyrgeometer) to gain deeper understanding of snow processes in this area. The possibility of sharing these data and discussing the outcomes of this research work with the snow community in INARCH opens a collaborative framework for new and wider opportunities to answer relevant scientific questions and share scientific knowledge from different mountain regions in the world. Moreover, the availability of the information from open access platforms offer other application and research fields validated data sets in research branches such as remote sensing of the snow pack but also ecology, water quality or fire risks, among others, widening the international community of potential users and multidisciplinary interactions.

**Conflicts of interest**

The authors declare that they have no conflict of interest.

**Special issue statement**

This article is part of the special issue "Hydrometeorological data from mountain and alpine research catchments". It is not associated with a conference.

**Acknowledgements**

The Guadalfeo Monitoring Network (GMN) is the extension of the initial weather station at Refugio Poqueira in 2004 within the framework of the Guadalfeo Project (Andalusian Regional Government), coordinated by Prof. Miguel A. Losada from the Andalusian Institute for Earth System Research at the University of Granada, whose continuous support and inspiration made it possible the creation of the Fluvial Dynamics and Hydrology research group in 2009, responsible of the GMN ever since.

The authors thank both J.I. López-Moreno and S. Gascoin for their comments and contribution to the final version of this manuscript.

This study was funded by the following research projects funded by the Spanish Ministry of Economy and Development (MINECO): Research Project CGL 2014-58508R, "Global monitoring system for snow areas in Mediterranean regions: trends analysis and implications for water resource management in Sierra Nevada", and Research Project CGL 2011-25632, "Snow dynamics in Mediterranean regions and its modelling at different scales. Implication for water management"; and cofinanced by the European Regional Development Fund (ERFD); the annual editions of the Research Program of the University of Cordoba also contributed to the instrumentation renewal and maintaining.

The authors want to acknowledge the International Network for Alpine Research Catchment Hydrology (INARCH) for the opportunity to share individual research experiences and data. The continuous support of the Natural and National Park of Sierra Nevada is also determinant for the development of this line of research since 2002.

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
