# Peer review of "The Guadalfeo Monitoring Network (Sierra Nevada, Spain): 14 years of measurements to understand the complexity of snow dynamics in semiarid regions"

_Earth System Science Data, 2018_

## Referee Comment (RC1) · López-Moreno (Referee) · 27 Nov 2018

The submitted papers deals with an area really interesting as it is one of the best Mediterranean places to study snow and snow hydrology. The manuscript is very well written and it is not only interesting for the dataset offered but also for the section of applications with the data displayed in the manuscript. Definetively, I think this will be a very nice contribution for the special issue. I have only two comments that authors and editor should consider for preparing a final version of the manuscript: 1- The most important is the daily nature of provided data. I think the value of the data

set would definitively improve if authors provide subdaily data (10 minutes-hourly) as it could permit to be used to run different simulations or to validate atmospheric products at subdaily resolution. This fits much better with the type of experiments aimed in INARCH project, and this is what the majority of the contributing papers to the special issue are doing. 2- I would write a little more about possible uncertainty in the data and which procedure (if any) has been used for quality control and gap filling. I also wonder if the solid precipitation data contains some correction for undercatch. If not some comment (or data) about wind speeds during snowfall events could help to get an idea about the importance of this process in Sierra Nevada.

Other minor comments: P3l28: "This main orientation...." I do not understand very well this sentence" P3L33: It sound to me the sentence a bit awkward.. P4l4-5. HM3..3 as superscript May be Fig 1A and Fig 2 could be merged F6, I would show the full name of the two snow variables. H is not very conventional, may be better HS (height of snow), and make consistent the name with table 5. I would improve the labels of Y-axis in figure 7.

There is only provided data from the camera installed in Poqueira, I wonder if it is worth to explain and show in the manuscript the other two cameras.

Hoping my comments will be useful, Ignacio López

---

## Referee Comment (RC2) · Gascoin (Referee) · 25 Jan 2019

This paper introduces two data sets to study snow cover dynamics in the Sierra Nevada, Spain: (i) meteorological data from five automatic weather stations since 2009 (ii) fractional snow cover area and snow depth at two (to be checked - see below) high elevation sites from time lapse cameras.

Many semi-arid and Mediterranean regions rely on snowmelt for water resources supply. However, there are few monitoring networks like this one in semi-arid mountain

[Figure]

regions. As a result there is often a "wet bias" in the evaluation and development of snow and hydrological models. Therefore, the publication of the Guadalfeo monitoring network data should be applauded. In addition, the records have almost no gaps which denotes the careful maintenance of the stations over these years (Fig. 3), despite the remote location of some stations.

The paper reads well and I have only two major comments:

- I concur with the first referee that the meteorological data should be provided at the hourly time step at least (currently only daily data are available in Pangaea). This is important because an objective of this special issue is to gather evaluation data for atmospheric circulation model in complex terrain and specifically their ability to resolve the diurnal cycle of surface level meteorological variables. In addition, sub-daily fluctuations of wind, radiation, temperature and humidity are required to run an energy balance snowpack model. Last, daily air temperatures does not allow an accurate determination of the precipitation phase (snow vs. rain). I also agree that the authors should make clear if a snow undercatch correction was applied.

- In the second repository there is only snow depth and snow fraction from camera C2 (https://doi.org/10.1594/PANGAEA.871706), whereas the abstract states that data from two time-lapse cameras are provided. In the main text three time-lapse cameras are presented (e.g. Tab. 2). Maybe this is a mistake but I encourage the authors to share all the data from cameras C1 and C3 since they have a much larger coverage than C2 (which covers only a plot of 30 m by 30 m). In addition I encourage the authors to share the snow cover *maps* (Fig. 5) and not only the time series of the average snow fraction. This would be very useful for the evaluation of remote sensing products. If the authors do not want to share the snow cover maps then at least a shapefile of the imaged area should be provided for each camera.

_Minor comments_

P1L24: cause, not show

[Figure]

P1L27: any reference to justify this statement?

P2L5: eastern rather that western Pyrenees

P2L6: Mount Lebanon and Anti Lebanon (or Lebanese mountain ranges)

P2L7: a "snow" paper could be cited for each mountain range (see for instance a review by Fayad et al. 2017 in J. Hydrol.)

P2L9: laboratories

P2L9: have, not having

P2L17: I am wondering if we really state that the spatial distribution of snow (what variable by the way?) is more variable is semi-arid regions? Large snow depth variability is also found in temperate alpine regions, but it may be less "visible" than in areas of shallow snowpacks.

P4L5: it is a detail but I do not understand the rationale of this sentence: if the snow influence is damped it should be less interesting for snow studies?

P4L14: specify from which station (or is it from a model run?) this average was computed.

P9L3: it would be useful to indicate the accuracy of the snow depth and snow fraction from the camera data.

P15, Fig 7: how were snow and rain separated from the total precipitation?

P17, Fig 8: can you explain to what conditions relate each "curve"?

I hope my comments will be useful, best regards.

---

## Author Comment (AC1) · 22 Feb 2019

We would like to acknowledge and thank the Reviewers and Editor for their work and useful and interesting comments, which was really helpful to improve the manuscript. We are attaching 3 documents:

- 1_Response to Reviewers.pdf: a point-by-point answer to both Reviewers, in which reference to modifications in the paper is included when needed as lines-pages in the revised manuscript (also attached).

[Figure]

- 2_Revised_Manuscript.pdf: revised version of the manuscript (removed text crossed out and new text in red).

- 3_New_Manuscript.pdf: new version of the manuscript.

Regards

/The authors

Please also note the supplement to this comment:
https://www.earth-syst-sci-data-discuss.net/essd-2018-123/essd-2018-123-AC1-supplement.zip

————————————————————

---

## Author Response (AR1)

*We would like to acknowledge and thank the Reviewers and Editor for their work and useful and interesting comments, which was really helpful to improve the manuscript. Please, find below a point-by-point answer to both Reviewers, in which reference to modifications in the paper is included when needed as lines-pages in the revised manuscript (also attached).*

**Reviewer 1**

**The submitted papers deals with an area really interesting as it is one of the best Mediterranean places to study snow and snow hydrology. The manuscript is very well written and it is not only interesting for the dataset offered but also for the section of applications with the data displayed in the manuscript. Definitely, I think this will be a very nice contribution for the special issue.**

*We would like to thank Reviewer #1 very much for his appreciation.*

**I have only two comments that authors and editor should consider for preparing a final version of the manuscript:**

**1- The most important is the daily nature of provided data. I think the value of the data set would definitively improve if authors provide subdaily data (10 minutes-hourly) as it could permit to be used to run different simulations or to validate atmospheric products at subdaily resolution. This fits much better with the type of experiments aimed in INARCH project, and this is what the majority of the contributing papers to the special issue are doing.**

*Following this comment, we have included hourly series of the data in the repository and modified the text accordingly (see page 20 lines 10-11 in the revised manuscript). Please, find the new datasets accessible from*
**https://doi.org/10.1594/PANGAEA.895236**

**2- I would write a little more about possible uncertainty in the data and which procedure (if any) has been used for quality control and gap filling. I also wonder if the solid precipitation data contains some correction for undercatch. If not some comment (or data) about wind speeds during snowfall events could help to get an idea about the importance of this process in Sierra Nevada.**

All data undergo a quality control protocol before being delivered that consists of a standard limit checking, intercomparison of values for each variable with the nearby stations, and internal checking of the values of a given variable in the context of the whole set of information at the same station (including video image). As a result, any suspicious 5-min record is removed from the series; to generate hourly series, only periods with less than two removals are kept (i.e. a maximum uncertainty of 8.33% is due to gaps in the record), and only days with complete hourly values are included in the daily series. No gap filling is done on the resulting series on any time scale to keep the original data set, and it is up to the user to apply any gap filling technique if required. Additionally, periodic calibration of the solar radiation, temperature and relative humidity

sensors is performed on an annual basis by means of portable calibrated sensors provisionally installed beside each station. The fraction of removed records varies depending on the variable, but is generally below 0.01% in the network.

Regarding undercatch, no correction is done on the data; the pluviometers above 1500 m a.s.l. are equipped with either Alter or Tretiakov shields to improve snow catching; the different field tests performed at the stations sites showed negligible undercatch amounts during the most frequent events. The mean value of wind speed during snowfall events ranges between 3.1 and 4.8 m/s at the stations PG1 to PG4 (with mode values ranging from 1 to 2.5 m/s), and 0.8 m/s at station PG5 (with mode value of 1 m/s). However, under extreme conditions this might not hold; during the study period the maximum wind speed registered ranged from 12.1 m/s (PG3) and 24.7 m/s (PG1). This has been explicitly clarified in the revised text (see page 8 lines 19-26 in the revised manuscript). Again, we deliver the data without any modification so that the interested users can decide whether correct or not for their own applications, provided that the wind data are also included in the dataset.

Following this comment, new paragraphs have been included to clarify these issues in the text (see page 8, lines 10-26 in the revised manuscript).

**Other minor comments:**

**P3I28: "This main orientation...." I do not understand very well this sentence"**
We have rewritten the sentence for better explanation (see page 4 line 2 in the revised version).

**P3L33: It sound to me the sentence a bit awkward..**
Yes, it really does, we apologize for this. We have rewritten the sentence (see page 4 lines 6-7 in the revised version).

**P4I4-5. HM3..3 as superscript**
We have corrected this typo.

**May be Fig 1A and Fig 2 could be merged**
Each figure illustrates different aspects although they share the basin. While Figure 1 provides us with a general overview of the whole mountainous range, and shows the Guadalfeo Basin and some pictures that help to understand the landscape of the network, Figure 2 is focused on the elements of the monitoring network (i.e. location of the meteorological stations, times-lapse cameras, streamflow gauge stations) in the basin in the context of other existing stations from public networks. We would like to keep both figures in the final version, if accepted, since merging would make it difficult to include all the points in a visible style.

**F6, I would show the full name of the two snow variables. H is not very conventional, may be better HS (height of snow), and make consistent the name with table 5.**

We have replaced h for HS in Fig. 6 and written the full name of the variables in both the figure caption and Table 3.

**I would improve the labels of Y-axis in figure 7.**
We have increased the font size of the Y-axis labels in Fig. 7 in the revised version.

**There is only provided data from the camera installed in Poqueira, I wonder if it is worth to explain and show in the manuscript the other two cameras.**

The images series from the camera C2 in PG2 is the most complete in the network: it is the longest, and it was taken by the same camara during all the period; camera C1 had some gap intervals and was replaced during the study period, and C3's scene includes two different horizons and is affected by shadowing, both of them requiring a special treatment to provide a continuous and coherent series of snow cover maps, which has not been completed so far. That is the reason why only the dataset from C2 has been selected in this work.

However, we had already processed those images from C1 that overlap with the available cloud-free images from Landsat TM sensors in the context of the work by Pimentel et al. (2017), cited in the text, in which validation of a spectral mixture model to retrieve snow cover area from this data source was done by using the images as ground-truth data set. Following this comment and the suggestion of Reviewer #2 we have included such maps in the data sets associated to the paper; they are accessible from:

**https://doi.pangaea.de/10.1594/PANGAEA.898374**

**Hoping my comments will be useful, Ignacio López**

**Reviewer 2**

**This paper introduces two data sets to study snow cover dynamics in the Sierra Nevada, Spain: (i) meteorological data from five automatic weather stations since 2009 (ii) fractional snow cover area and snow depth at two (to be checked - see below) high elevation sites from time lapse cameras. Many semi-arid and Mediterranean regions rely on snowmelt for water resources supply. However, there are few monitoring networks like this one in semi-arid mountain regions. As a result there is often a "wet bias" in the evaluation and development of snow and hydrological models. Therefore, the publication of the Guadalfeo monitoring network data should be applauded. In addition, the records have almost no gaps which denotes the careful maintenance of the stations over these years (Fig. 3), despite the remote location of some stations.**

*We would like to thank Reviewer #2 very much for his appreciation.*

**The paper reads well and I have only two major comments:**

**- I concur with the first referee that the meteorological data should be provided at the hourly time step at least (currently only daily data are available in Pangaea). This is important because an objective of this special issue is to gather evaluation data for atmospheric circulation model in complex terrain and specifically their ability to resolve the diurnal cycle of surface level meteorological variables. In addition, sub-daily fluctuations of wind, radiation, temperature and humidity are required to run an energy balance snowpack model. Last, daily air temperatures does not allow an accurate determination of the precipitation phase (snow vs. rain). I also agree that the authors should make clear if a snow undercatch correction was applied.**

**- In the second repository there is only snow depth and snow fraction from camera C2 (https://doi.org/10.1594/PANGAEA.871706), whereas the abstract states that data from two time-lapse cameras are provided. In the main text three time-lapse cameras are presented (e.g. Tab. 2). Maybe this is a mistake but I encourage the authors to share all the data from cameras C1 and C3 since they have a much larger coverage than C2 (which covers only a plot of 30 m by 30 m). In addition I encourage the authors to share the snow cover \*maps\* (Fig. 5) and not only the time series of the average snow fraction. This would be very useful for the evaluation of remote sensing products. If the authors do not want to share the snow cover maps then at least a shapefile of the imaged area should be provided for each camera.**

Following this comment, as explained before, hourly data have been included to the repository. Regarding the data sets from the cameras, we have also explained the reasons why only C2 data sets are included in the work; but following this comment, the available snow maps obtained from C1, with the largest coverage, are now also available.

**https://doi.pangaea.de/10.1594/PANGAEA.898374**

Additionally to the new paragraphs in the text regarding these comments, we have modified the abstract accordingly. (see page 1 lines 17-20, and page 20 lines 10-16 in the revised manuscript).

**Minor comments**

**P1L24: cause, not show**

We have modified this in the revised text.

**P1L27: any reference to justify this statement?**

A new reference has been added:
*Musselman, K.N, Clark, M., Liu, C., Ikeda, K., Rasmussen, R.: Slower snowmelt in a warmer world, Nature Climate Change, 7, 214-119, doi: 10.1038/nclimate3225, 2017*

**P2L5: eastern rather that western Pyrenees**

We have modified this in the revised manuscript (see page 2, line 8)

**P2L6: Mount Lebanon and Anti Lebanon (or Lebanese mountain ranges)**

We have modified this in the revised manuscript (see page 2, line 11)

**P2L7: a "snow" paper could be cited for each mountain range (see for instance a review by Fayad et al. 2017 in J. Hydrol.)**

We have added new references, as suggested, for each cited mountain range in the revised text (see page 2 lines 8-12 in the revised manuscript) and in the References' section.

*Favier, V., Falvey, M., Rabatel, A., Praderio, E., López, D.: Interpreting discrepancies between discharge and precipitation in high-altitude area of Chile's Norte Chico region (26–32°S), Water Resour. Res., 45 (2), p. W02424, 2009.*

*Fayad, A., Gascoin, S., Faour, G., López-Moreno, J.I., Drapeau, L., Le Page, M., Escadafal, R.: Snow hydrology in Mediterranean mountain regions: A review, J. of Hydrol., 551, 374-396, 2017.*

*Lopez-Moreno, J.I., Goyette, S., Beniston, M.: Impact of climate change on snowpack in the Pyrenees: Horizontal spatial variability and vertical gradients, J. of Hydrol., 374 (3-4): 384-396, 2009*

*Marchane, A., Jarlan, L., Hanich, L., Boudhar, A., Gascoin, S.,Tavernier, A., Filali, N., le Page, M., Hagolle, O., Berjamy, B.: Assessment of daily MODIS snow cover products to monitor snow cover dynamics over the Moroccan Atlas mountain range, Remote Sens. Environ., 160, 72-86, 2015.*

*Mhawej, M., Faour, G., Fayad, A., Shaban, A.: Towards an enhanced method to map snow cover areas and derive snow-water equivalent in Lebanon, J. of Hydrol., 513, 274-282, 2014.*

*Molotch, N., Meromy, L.: Physiographic and climatic controls on snow cover persistence in the Sierra Nevada Mountains, Hydrol. Process, 28, 4573-4586, 2014.*

*Pérez-Palazón, M., Pimentel, R., Polo, M., Pérez-Palazón, M. J., Pimentel, R. and Polo, M. J.: Climate Trends Impact on the Snowfall Regime in Mediterranean Mountain Areas: Future Scenario Assessment in Sierra Nevada (Spain), Water, 10(6), 720, doi:10.3390/w10060720, 2018.*

*Senatore, A., Mendicino, G., Smiatek, G., Kunstmann, H.: Regional climate change projections and hydrological impact analysis for a Mediterranean basin in Southern Italy, J. Hydrol., 399, 70-92, 2011.*

**P2L9: laboratories**

We have modified this in the revised manuscript (see page 2, line 14)

**P2L9: have, not having**

We have modified this in the revised manuscript (see page 2, line 15)

**P2L17: I am wondering if we really state that the spatial distribution of snow (what variable by the way?) is more variable is semi-arid regions? Large snow depth variability is also found in temperate alpine regions, but it may be less "visible" than in areas of shallow snowpacks.**

This is, in fact, necessary to clarify in the text, thank you for this specific comment. We meant the spatial coverage of the snowpack in these areas, with a frequently patchy behaviour not only at the end of the snow season, but also during the fall and winter due to the different accumulation-ablation cycles that occur. We have explained this better in the revised text (see page 2 lines 19-21 in the revised manuscript).

**P4L5: it is a detail but I do not understand the rationale of this sentence: if the snow influence is damped it should be less interesting for snow studies?**

By this sentence we meant that the dam affects the natural flow regime downstream and the available records there cannot be used to close any balance equation without considering the reservoir dynamics, which is not simple nor straight forward. We select the dam location to close this experimental basin and use the available daily inflow records for water and energy balance closure. This sentence has been rewritten to avoid misunderstanding (see lines 11-13 in page 4 in the revised text).

**P4L14: specify from which station (or is it from a model run?) this average was computed.**

These are average values in the Guadalfeo Experimental Catchment described in the paper and Figure 1. These values are calculated over a reference period (1960-2000) with all the meteorological information available in the catchment for this period (see location of the station

as black dots in Fig. 2; Pérez-Palazón et al., 2015). We have added this information in the text and the reference (see page 4 lines 18-21 in the revised manuscript).

*Pérez-Palazón, M. J., Pimentel, R., Herrero, J., Aguilar, C., Perales, J. M., and Polo, M. J.: Extreme values of snow-related variables in Mediterranean regions: trends and long-term forecasting in Sierra Nevada (Spain), Proc. IAHS, 369, 157-162, https://doi.org/10.5194/piahs-369-157-2015, 2015.*

**P9L3: it would be useful to indicate the accuracy of the snow depth and snow fraction from the camera data.**

We have added new text following this comment (see lines 12-17 in page 10 in the revised text). The accuracy of the observations from the camera data was estimated as 0.075 $m^2\,m^{-2}$ for the snow cover fraction (SCF) and 50 mm for the height of snow (HS). The associated error for SCF is due to the combined effects of both the georeferencing process and the snow detection algorithm; the former depends on the relationship between the number of pixels in the images and the resolution of the local DEM, and the latter is related to the accuracy of the K-means algorithm used. In the case of HS, the error depends on the position of the rods installed in the control area and also on the accuracy of the clustering algorithm used.

**P15, Fig 7: how were snow and rain separated from the total precipitation?**

In Figure 7, the snowfall/rainfall classification is calculated on an hourly basis by using a threshold temperature of 0ºC at each station (see lines 18-19 in page 16 in the revised text).

**P17, Fig 8: can you explain to what conditions relate each "curve"?**

Yes, this was studied and discussed in the work by Pimentel et al. (2017) in HESS cited in the manuscript. We have included here a brief explanation (see lines 26-31 in page 18 in the revised manuscript).

[revised manuscript text omitted]

---

## Author Response (AR2)

Dear Editor,

Thank you very much for your nice appreciation and work. We are very happy with and acknowledge the benefit the feedback by Reviewers has had on our manuscript, and to be finally part of this relevant special issue in our research field.

We have changed the words pointed out in your file in the new version of the manuscript and uploaded this and the rest of required information to the ESSD platform. We are ready for any additional editing that should be required.

On behalf of all Authors, thank you again, with our best regards,

María J. Polo

[revised manuscript text omitted]